# SAFE REINFORCEMENT LEARNING WITH CONTRASTIVE RISK PREDICTION

## ABSTRACT

As safety violations can lead to severe consequences in real-world applications, the increasing deployment of Reinforcement Learning (RL) in safety-critical domains such as robotics has propelled the study of safe exploration for reinforcement learning (safe RL). In this work, we propose a risk preventive training method for safe RL, which learns a statistical contrastive classifier to predict the probability of a state-action pair leading to unsafe states. Based on the predicted risk probabilities, we can collect risk preventive trajectories and reshape the reward function with risk penalties to induce safe RL policies. We conduct experiments in robotic simulation environments. The results show the proposed approach has comparable performance with the state-of-the-art model-based methods and outperforms conventional model-free safe RL approaches.

## 1 INTRODUCTION

Reinforcement Learning (RL) offers a great set of technical tools for many real-world decision making systems, such as robotics, that require an agent to automatically learn behavior policies through interactions with the environments (Kober et al., 2013). Conversely, the applications of RL in real-world domains also pose important new challenges for RL research. In particular, many real-world robotic environments and tasks, such as human-related robotic environments (Brunke et al., 2021), helicopter manipulation (Martín H & Lope, 2009; Koppejan & Whiteson, 2011), autonomous vehicle (Wen et al., 2020), and aerial delivery (Faust et al., 2017), have very low tolerance for violations of safety constraints, as such violation can cause severe consequences. This raises a substantial demand for safe reinforcement learning techniques.

Safe exploration for RL (safe RL) investigates RL methodologies with critical safety considerations, and has received increased attention from the RL research community. In safe RL, in addition to the reward function (Sutton & Barto, 2018), an RL agent often deploys a cost function to maximize the discounted cumulative reward while satisfying the cost constraint (Mihatsch & Neuneier, 2002; Hans et al., 2008; Ma et al., 2022). A comprehensive survey of safe RL categorizes the safe RL techniques into two classes: modification of the optimality criterion and modification of the exploration process (García & Fernández, 2015). For modification of the optimality criterion, previous works mostly focus on the modification of the reward. Many works (Ray et al., 2019; Shen et al., 2022; Tessler et al., 2018; Hu et al., 2020; Thomas et al., 2021; Zhang et al., 2020) pursue such modifications by shaping the reward function with penalizations induced from different forms of cost constraints. For modification of the exploration process, safe RL approaches focus on training RL agents on modified trajectory data. For example, some works deploy backup policies to recover from safety violations to safer trajectory data that satisfy the safety constraint (Thananjeyan et al., 2021; Bastani et al., 2021; Achiam et al., 2017).

In this paper, we propose a novel risk preventive training (RPT) method to tackle the safe RL problem. The key idea is to learn a contrastive classification model to predict the risk—the probability of a state-action pair leading to unsafe states, which can then be deployed to modify both the exploration process and the optimality criterion. In terms of exploration process modification, we collect trajectory data in a risk preventive manner based on the predicted probability of risk. A trajectory is terminated if the next state falls into an unsafe region that has above-threshold risk values. Regarding optimality criterion modification, we reshape the reward function by penalizing it with the predicted risk for each state-action pair. Benefiting from the generalizability of risk prediction, the proposed

approach can avoid safety constraint violations much early in the training phase and induce safe RL policies, while previous works focus on backup policy and violate more safety constraints by interacting with the environment in the unsafe regions. We conduct experiments using four robotic simulation environments on MuJoCo (Todorov et al., 2012). Our model-free approach produces comparable performance with a state-of-the-art model-based safe RL method SMBPO (Thomas et al., 2021) and greatly outperforms other model-free safe RL methods. The main contributions of the proposed work can be summarized as follows:

- This is the first work that introduces a contrastive classifier to perform risk prediction and conduct safe RL exploration.
- With risk prediction probabilities, the proposed approach is able to perform both exploration process modification through risk preventive trajectory collection and optimality criterion modification through reward reshaping.
- As a model-free method, the proposed approach achieves comparable performance to the state-of-the-art model-based safe RL method and outperforms other model-free methods in robotic simulation environments.

## 2 RELATED WORKS

Many methods have been developed in the literature for safe RL. Altman (1999) first introduced the Constrained Markov Decision Process (CMDP) to formally define the problem of safe exploration in reinforcement learning. Mihatsch & Neuneier (2002) introduced the definition of risk for safe RL and intended to find a risk-avoiding policy based on risk-sensitive controls. Hans et al. (2008) further differentiated the states as "safe" and "unsafe" states based on human-designed criteria, while an RL agent is considered to be not safe if it reaches "unsafe" states. Garcıa & Fernández (2015) presented a comprehensive survey on safe RL, which categorizes safe RL methods into two classes: modification of the optimality criterion and modification of the exploration process.

**Modification of the optimality criterion.** Since the optimization of conventional criterion (long-term cumulative reward) does not ensure the avoidance of safety violations, previous works have studied the modification of the optimality objective, based on different notions of risk (Howard & Matheson, 1972; Sato et al., 2001), probabilities of visiting risky states (Geibel & Wysotzki, 2005), etc. Achiam et al. (2017) proposed a Constrained Policy Optimization (CPO) to update the safe policy by optimizing the primal-dual problem in trust regions. Recently, reward shaping (Dorigo & Colombetti, 1994; Randløv & Alstrøm, 1998) techniques have been brought into the safe exploration of RL. Tessler et al. (2018) applied the reward shaping technique in safe RL to penalize the normal training policy, which is known as Reward Constrained Policy Optimization (RCPO). Zhang et al. (2020) developed a reward shaping approach built upon Probabilistic Ensembles with Trajectory Sampling (PETS) (Chua et al., 2018) that maximizes the average return of predicted horizons. It pre-trains an predictor of the unsafe state in an offline sandbox environment and penalizes the reward of PETS during the adaptation in online environments. A similar work in (Thomas et al., 2021) reshapes reward functions using a model-based predictor. It regards unsafe states as absorbing states and trains the RL agent with a penalized reward to avoid the visited unsafe states.

**Modification of the exploration process.** Some previous works have attempted to optimize the safe RL policy by interacting with the environment with adjusted exploration processes. For example, Driessens & Džeroski (2004); Martín H & Lope (2009); Song et al. (2012) provided guidance to the exploration process based on prior knowledge on the environment. Similarly, Abbeel et al. (2010); Tang et al. (2010) restricted the exploration process learning based on demonstration data. More recently, Thananjeyan et al. (2021); Bastani et al. (2021) focused on using backup policies of the safe regions, aiming to avoid safety violations. If the agent takes a potentially dangerous action, the task policy will be replaced with a guaranteed safe backup policy. Ma et al. (2022) proposed a model-based conservative and adaptive penalty approach to explore safely by modifying the penalty adaptively in the training process.

Safe RL is important for application environments with limited cost for trial-and-error, such as the human-related robotic environments, where violations of safety concerns may lead to catastrophic failures (Brunke et al., 2021). Todorov et al. (2012) developed a robotic simulation environment

named MuJoCo, which promotes the study of the RL applications in robotic environments. Thomas et al. (2021) further modified the MuJoCo environment to define safety violations for robotic simulations. Such environments provided a suitable test bed for safe RL methods.

## 3 PRELIMINARY

Reinforcement learning (RL) has been broadly used to train robotic agents by maximizing the discounted cumulative rewards. The representation of a reinforcement learning problem can be formulated as a Markov Decision Process (MDP) $M = (\mathcal{S}, \mathcal{A}, \mathcal{T}, \mathcal{R}, \gamma)$ (Sutton & Barto, 2018), where $\mathcal{S}$ is the state space for all observations, $\mathcal{A}$ is the action space for available actions, $\mathcal{T} : \mathcal{S} \times \mathcal{A} \to \mathcal{S}$ is the transition dynamics, $\mathcal{R} : \mathcal{S} \times \mathcal{A} \to [r_{min}, r_{max}]$ is the reward function, and $\gamma \in (0, 1)$ is the discount factor. An agent can start from a random initial state $s_0$ to take actions and interact with the MDP environment by receiving rewards for each action and moving to new states. Such interactions can produce a transition $(s_t, a_t, r_t, s_{t+1})$ at each time-step $t$ with $s_{t+1} = \mathcal{T}(s_t, a_t)$ and $r_t = c(s_t, a_t)$, while a sequence of transitions comprise a trajectory $\tau = (s_0, a_0, r_0, s_1, a_1, r_1, \cdots, s_{|\tau|+1})$, where $|\tau| + 1$ denotes the length of trajectory $\tau$—i.e., the number of transitions. The goal of RL is to learn an optimal policy $\pi^\star : \mathcal{S} \to \mathcal{A}$ that can maximize the expected discounted cumulative reward (return): $\pi^\star = \arg\max_\pi \ J_r(\pi) = \mathbb{E}_{\tau \sim \mathcal{D}_\pi}[\sum_{t=0}^{|\tau|} \gamma^t r_t]$

### 3.1 SAFE EXPLORATION FOR REINFORCEMENT LEARNING

Safe exploration for Reinforcement Learning (safe RL) studies RL with critical safety considerations. For a safe RL environment, in addition to the reward function, a cost function can also exist to reflect the risky status of each exploration step. The process of safe RL can be formulated as a Constrained Markov Decision Process (CMDP) (Altman, 1999), $\hat{M} = (\mathcal{S}, \mathcal{A}, \mathcal{T}, \mathcal{R}, \gamma, c, d)$, which introduces an extra cost function $c$ and a cost threshold $d$ into MDP. An exploration trajectory under CMDP can be written as $\tau = (s_0, a_0, r_0, c_0, s_1, a_1, r_1, c_1, \cdots, s_{|\tau|+1})$, where the transition at time-step $t$ is $(s_t, a_t, s_{t+1}, r_t, c_t)$, with a cost value $c_t$ induced from the cost function $c_t = c(s_t, a_t)$. CMDP monitors the safe exploration process by requiring the cumulative cost $J_c(\pi)$ does not exceed the cost threshold $d$, where $J_c(\pi)$ can be defined as the expected total cost of the exploration, $J_c(\pi) = \mathbb{E}_{\tau \sim \mathcal{D}_\pi}[\sum_{t=0}^{|\tau|} c_t]$ (Ray et al., 2019). Safe RL hence aims to learn an optimal policy $\pi^\star$ that can maximize the expected discounted cumulative reward subjecting to a cost constraint, as follows:

$$\pi^\star = \arg\max_\pi J_r(\pi) = \mathbb{E}_{\tau \sim \mathcal{D}_\pi}\left[\sum_{t=0}^{|\tau|} \gamma^t r_t\right], \quad \text{s.t. } J_c(\pi) = \mathbb{E}_{\tau \sim \mathcal{D}_\pi}\left[\sum_{t=0}^{|\tau|} c_t\right] \leq d. \quad (1)$$

## 4 METHOD

Robot operations typically have low tolerance for risky/unsafe states and actions, since a robot could be severely damaged in real-world environments when the safety constraint being violated. Similar to the work in (Hans et al., 2008), we adopt a strict setting in this work for the safety constraint such that any "unsafe" state can cause violation of the safety constraint and the RL agent will terminate an exploration trajectory when encountering an "unsafe" state. We have the following definition:

**Definition 1.** *For a state $s$ and an action $a$, the value of the cost function $c(s, a)$ can either be $0$ or $1$. When $c(s, a) = 0$, the induced state $\mathcal{T}(s, a)$ is defined as a safe state; when $c(s, a) = 1$, the induced state $\mathcal{T}(s, a)$ is defined as an unsafe state, which triggers the violation of safety constraint and hence causes the termination of the trajectory.*

Based on this definition, the cost threshold $d$ in Eq. (1) should be set strictly to $0$. The agent is expected to learn a safe policy $\pi$ that can operate with successful trajectories containing only safe states. Towards this goal, we propose a novel risk prediction method for safe RL. The proposed method deploys a contrastive classifier to predict the probability of a state-action pair leading to unsafe states, which can be trained during the exploration process of RL and generalized to previously unseen states. With risk prediction probabilities, a more informative cumulative cost $J_c(\pi)$ can be formed to prevent unsafe trajectories and reshape the reward in each transition of a trajectory to induce safe RL policies. Previous safe RL methods in the literature can typically be categories into two classes: modification of the optimality criterion and modification of the exploration process (García

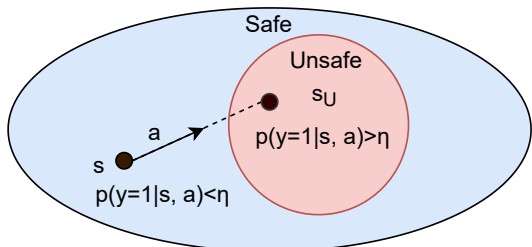

Figure 1: An overview of the risk preventive process. The blue area denotes the safe region of the trajectory. The dot $s_U$ is an unsafe state. The red area around the unsafe state $s_U$ denotes the unsafe region, bounded by the value of risk prediction $p(y = 1|s, a)$. The RL agent explores risk preventive trajectories by avoiding entering the unsafe region in the training process.

& Fernández, 2015). With safety constraints and risk predictions, the proposed approach (to be elaborated below) is expected to incorporate the strengths of both categories of safe RL techniques.

### 4.1 RISK PREDICTION WITH CONTRASTIVE CLASSIFICATION

Although an RL agent would inevitably encounter unsafe states during the initial stage of the exploration process in an unknown environment, we aim to quickly learn from the unsafe experience through statistical learning and generalize the recognition of unsafe trajectories to prevent risk for future exploration. Specifically, we aim to compute the probability of a state-action pair leading to unsafe states, i.e., $p(y = 1|s_t, a_t)$, where $y \in \{0, 1\}$ denotes a random variable that indicates whether $(s_t, a_t)$ leads to an unsafe state $s_u \in S_U$. The set of unsafe states, $S_U$, can be either pre-given or collected during initial exploration. However, directly training a binary classifier to make such predictions is impractical as it is difficult to judge whether a state-action pair is *safe*—i.e., never leading to unsafe states. For this purpose, we propose to train a contrastive classifier $F_\theta(s_t, a_t)$ with model parameter $\theta$ to discriminate a positive state-action pair $(s_t, a_t)$ in a trajectory that leads to unsafe states (unsafe trajectory) and a random state-action pair from the overall distribution of any trajectory. Such a contrastive form of learning can conveniently avoid the identification problem of absolute negative (safe) state-action pairs.

Let $p(s_t, a_t|y = 1)$ denote the presence probability of a state-action pair $(s_t, a_t)$ in a trajectory that leads to unsafe states, and $p(y = 1)$ denote the distribution probability of unsafe trajectory in the environment. The contrastive classifier $F_\theta(s_t, a_t)$ is defined as follows:

$$F_\theta(s_t, a_t) = \frac{p(s_t, a_t|y = 1)p(y = 1)}{p(s_t, a_t|y = 1)p(y = 1) + p(s_t, a_t)}, \tag{2}$$

where $p(y = 1)$ is used as weight for the positive samples and weight 1 is given to the *contrastively-negative* samples. This binary classifier identifies the state-action pairs in unsafe trajectories contrastively from general pairs in the overall distribution.

From the definition of $F_\theta(s_t, a_t)$ in Eq.(2), one can easily derive the probability of interest, $p(y = 1|s_t, a_t)$, by Bayes' theorem, as follows:

$$p(y = 1|s_t, a_t) = \frac{p(s_t, a_t|y = 1)p(y = 1)}{p(s_t, a_t)} = \frac{F_\theta(s_t, a_t)}{1 - F_\theta(s_t, a_t)}. \tag{3}$$

Although the normal output range for the probabilistic classifier $F_\theta(s_t, a_t)$ should be $[0, 1]$, this could lead to unbounded $p(y = 1|s_t, a_t) \in [0, \infty]$ through Eq.(3). Hence we propose to rescale the output of classifier $F_\theta(s_t, a_t)$ to the range of $[0, 0.5]$.

We optimize the contrastive classifier's parameter $\theta$ using maximum likelihood estimation (MLE). The log-likelihood objective function can be written as:

$$L(\theta) = p(y = 1)\mathbb{E}_{p(s_t, a_t|y=1)}[\log F_\theta(s_t, a_t)] + \mathbb{E}_{p(s_t, a_t)}[\log(1 - F_\theta(s_t, a_t))]. \tag{4}$$

### 4.2 RISK PREVENTIVE TRAJECTORY

Based on Definition 1, a trajectory terminates when the RL agent encounters an unsafe state and triggers safety constraint violation. It is however desirable to minimize the number of such safety

violations even during the policy training process and learn a good policy in safe regions. The risk prediction classifier we proposed above provides a convenient tool for this purpose by predicting the probability of a state-action pair leading to unsafe states, $p(y = 1|s_t, a_t)$. Based on this risk prediction, we have the following definition for unsafe regions:

**Definition 2.** *A state-action pair* $(s_t, a_t)$ *falls into an **unsafe region** if the probability of* $(s_t, a_t)$ *leading to unsafe states is greater than a threshold* $\eta$*:* $p(y = 1|s_t, a_t) > \eta$*, where* $\eta \in (0, 1)$*.*

With this definition, an RL agent can pursue risk preventive trajectories to avoid safety violations by staying away from unsafe regions. Specifically, we can terminate a trajectory before violating the safety constraint by judging the risk—the probability of $p(y = 1|s_t, a_t)$. The process is illustrated in Figure 1.

Without a doubt, the threshold $\eta$ is a key for determining the length $T = |\hat{\tau}|$ of an early stopped risk preventive trajectory $\hat{\tau}$. To approximate a derivable relation between $\eta$ and $T$, we make the following assumption and lemma:

**Assumption 1.** *For a trajectory* $\tau = \{s_0, a_0, r_0, c_0, s_1, \cdots, s_H\}$ *that leads to an unsafe state* $s_H \in S_U$*, the risk prediction probability* $p(y = 1|s_t, a_t)$ *increases linearly along the time steps.*

**Lemma 1.** *Assume that Assumption 1 holds. Let* $H \in \mathbb{N}$ *denote the length of an unsafe trajectory* $\tau = \{s_0, a_0, r_0, c_0, s_1, a_1, r_1, c_1, \cdots, s_H\}$ *that terminates at an unsafe state* $s_H \in S_U$*. The number of transition steps, $T$, along this trajectory to the unsafe region determined by $\eta$ in Definition 2 can be approximated as:* $T = \lfloor \frac{\eta - p_0}{1 - p_0} H \rfloor$*, where* $p_0 = p(y = 1|s_0, a_0)$*.*

*Proof.* According to assumption 1, the probability for $(s_t, a_t)$ leading to unsafe states $p(y_t = 1|s_t, a_t)$ increases linearly along time-steps. Let $p_0$ denote the probability of starting from the initial state $s_0$: $p_0 = p(y_t = 1|s_0, a_0)$. For a probability threshold $\eta \in (0, 1)$ for unsafe region identification in Definition 2, the ratio between the number of environment transition steps $T$ to the unsafe region and the unsafe trajectory length $H$ will be approximately (due to integer requirements over $T$) equal to the ratio between the probability differences of $\eta - p_0$ and $1 - p_0$. That is, $\frac{T}{H} \approx \frac{\eta - p_0}{1 - p_0}$. Hence $T$ can be approximated as: $T = \lfloor \frac{\eta - p_0}{1 - p_0} H \rfloor$. $\qquad\square$

This Lemma clearly indicates that a larger $\eta$ value will allow more effective explorations with longer trajectories, but also tighten the unsafe region and increase the possibility of safety violations.

### 4.3 RISK PREVENTIVE REWARD SHAPING

With Definition 1, the safe RL formulation in Eq. (1) can hardly induce a safe policy since there are no intermediate costs before encountering an unsafe state. With the risk prediction classifier proposed above, we can rectify this drawback by defining the cumulative cost function $J_c(\pi)$ using the risk prediction probabilities, $p(y = 1|s_t, a_t)$, over all encountered state-action pairs. Specifically, we adopt a reward-like discounted cumulative cost as follows: $J_c(\pi) = \mathbb{E}_{\tau \sim \mathcal{D}_\pi} \left[ \sum_{t=0}^{|\tau|} \gamma^t p(y = 1|s_t, a_t) \right]$, which uses the predicted risk at each time-step as the estimate cost. Moreover, instead of solving safe RL as a constrained discounted cumulative reward maximization problem, we propose using Lagrangian relaxation (Bertsekas, 1997) to convert the constrained maximization CMDP problem in Eq. (1) to an unconstrained optimization problem, which is equivalent to shaping the reward function with risk penalties:

$$\min_{\lambda \geq 0} \max_{\pi} \quad [J_r(\pi) - \lambda(J_c(\pi) - d)] \tag{5}$$

$$\Longleftrightarrow \quad \min_{\lambda \geq 0} \max_{\pi} \quad [J_r(\pi) - \lambda J_c(\pi)] \tag{6}$$

$$\Longleftrightarrow \quad \min_{\lambda \geq 0} \max_{\pi} \quad \mathbb{E}_{\tau \sim \mathcal{D}_\pi} \left[ \sum_{t=0}^{|\tau|} \gamma^t (r_t - \lambda p(y = 1|s_t, a_t)) \right]. \tag{7}$$

The Lagrangian dual variable $\lambda$ controls the degree of reward shaping with the predicted risk value.

**Theorem 1.** *To prevent the RL agent from falling into known unsafe states, the penalty factor (i.e., the dual variable) $\lambda$ for the shaped reward $\hat{r}_t = r_t - \lambda p(y_t = 1|s_t, a_t)$ should have the following*

*lower bound, where $H$ and $\eta$ are same as in Lemma 1:*

$$\lambda > \frac{(1 - \gamma^H)(r_{max} - r_{min})}{\eta\gamma^{\lfloor \frac{\eta - p_0}{1 - p_0} H \rfloor}\left(1 - \gamma^{H - \lfloor \frac{\eta - p_0}{1 - p_0} H \rfloor}\right)}. \tag{8}$$

*Proof.* For a trajectory with length $H$ that leads to an unsafe state $s_H \in S_U$, the largest penalized return of the unsafe trajectory should be smaller than the lowest possible unpenalized return from any safe trajectory with the same length:

$$\sup_\tau \left[\sum_{t=0}^{H-1} \gamma^t(r - \lambda p_t)\right] < \inf_\tau \left[\sum_{t=0}^{H-1} \gamma^t r\right], \tag{9}$$

where $p_t = p(y = 1|s_t, a_t)$. With this requirement, the RL agent can learn a policy to explore safe states and prevent the agent from entering unsafe states $S_U$ through unsafe regions. As the reward function $r$ is bounded within $[r_{min}, r_{max}]$, we can further simplify the inequality in Eq. (9) by seeking the supremum of its left-hand side (LHS) and the infimum of its right-hand side (RHS).

Based on Definition 2 and Lemma 1, we can split an unsafe trajectory $\tau$ with length $H$ into two sub-trajectories: a sub-trajectory within the safe region, $\tau_1 = (s_0, a_0, r_0, c_0, \cdots, s_{T-1}, a_{T-1}, r_{T-1}, c_{T-1})$, where $p_t \leq \eta$ and a sub-trajectory $\tau_2 = (s_T, a_T, r_T, c_T, \cdots, s_H)$ after entering the unsafe region determined by $p_t > \eta$. With Assumption 1, we have $p_t > \eta$ for all state-action pairs in the sub-trajectory $\tau_2$. Then the LHS and RHS of Eq. (9) can be upper bounded and lower bounded respectively as follows:

$$LHS \leq \sum_{t=0}^{H-1} \gamma^t r_{max} - \left(\sum_{t=0}^{T-1} \gamma^t \lambda \cdot 0 + \sum_{t=T}^{H-1} \gamma^t \lambda \eta\right), \tag{10}$$

$$RHS \geq \sum_{t=0}^{H-1} \gamma^t r_{min}. \tag{11}$$

To ensure the satisfaction of the requirement in Eq. (9), we then enforce the follows:

$$\sum_{t=0}^{H-1} \gamma^t r_{max} - \left(\sum_{t=0}^{T-1} \gamma^t \lambda \cdot 0 + \sum_{t=T}^{H-1} \gamma^t \lambda \eta\right) < \sum_{t=0}^{H-1} \gamma^t r_{min} \tag{12}$$

$$\iff \frac{(1 - \gamma^H)r_{max}}{1 - \gamma} - \frac{\lambda\eta \cdot \gamma^T(1 - \gamma^{H-T})}{1 - \gamma} < \frac{(1 - \gamma^H)r_{min}}{1 - \gamma} \tag{13}$$

$$\iff \lambda > \frac{(1 - \gamma^H)(r_{max} - r_{min})}{\eta\gamma^T(1 - \gamma^{H-T})}. \tag{14}$$

With Assumption 1 and Lemma 1, we can estimate the length $T$ for the safe sub-trajectory as $T = \lfloor \frac{\eta - p_0}{1 - p_0} H \rfloor$. With this estimation, the lower bound for $\lambda$ in Eq. (8) can be derived. $\qquad \square$

## 4.4 RISK PREVENTIVE TRAINING ALGORITHM

Our overall risk preventive RL training procedure is presented in Algorithm 1, which simultaneously trains the risk prediction classifier and performs reinforcement learning with risk preventive trajectory exploration (line 12-13) and risk preventive reward shaping (line 10).

## 5 EXPERIMENT

### 5.1 EXPERIMENTAL SETTINGS

**Experimental Environments** Following the experimental setting in (Thomas et al., 2021), we adopted four robotics simulation environments, *Ant, Cheetah, Hopper,* and *Humanoid*, using the MuJoCo simulator (Todorov et al., 2012). For *Ant* and *Hopper*, a robot violates the safety constraint when it falls over. For *Cheetah*, a robot violates the safety constraint when its head flips on the ground, which is modified from the HalfCheetah environment with extra safety constraint (Thomas et al., 2021). For *Humanoid*, the human-like robot violates the safety constraint when the head of the robot falls to the ground. The RL agent reaches the end of the trajectory once it encounters the safety violation.

---

**Algorithm 1** Risk Preventive Training

**Input** Initial policy $\pi_\phi$, classifier $F_\theta$, trajectory set $D = \emptyset$, set of unsafe state-action pairs $S_U$,
       threshold $\eta$,   penalty factor $\lambda$,   set of unsafe trajectory length $\mathcal{H} = \emptyset$

**Output** Trained policy $\pi_\phi$

1: **for** $k = 1, 2, ..., K$ **do**
2:     **for** $t = 1, 2, ..., T_{\max}$ **do**
3:         Sample transition $(s_t, a_t, r_t, c_t, s_{t+1})$ from the environment with policy $\pi_\phi$.
4:         **if** $c_t > 0$ **then**
5:             Add risky state-action $(s_t, a_t)$ into the unsafe state set $S_U$.
6:             Add length $t$ to $\mathcal{H}$. Increase $\lambda$ if the lower bound increases with $H = t$ and Eq. (8).
7:             Stop trajectory and break.
8:         Sample next action $a_{t+1}$ based on policy $\pi_\phi$ and next state $s_{t+1}$: $a_{t+1} = \pi_\phi(\cdot|s_{t+1})$.
9:         Calculate $p_t$ and $p_{t+1}$ using Eq. (3)
10:        Penalize reward $r_t$ with $p_t$:   $\hat{r}_t = r_t - \lambda p_t$
11:        Add transition to the trajectory set: $D = D \cup (s_t, a_t, \hat{r}_t, s_{t+1})$
12:        **if** $p_{t+1} > \eta$ **then**
13:           Stop trajectory and break.
14:     Sample risky state-action pairs from $S_U$
15:     Sample transitions from history: $(s_t, a_t, \hat{r}_t, s_{t+1}) \sim D$
16:     Update classifier $F_\theta$ by maximizing the likelihood $L(\theta)$ in Eq (4)
17:     Update policy $\pi_\phi$ with the shaped rewards $J_{\hat{r}}(\pi)$ in Eq (7)

---

**Comparison Methods**   We compare our proposed Risk Preventive Training (RPT) approach with three state-of-the-art safe RL methods: SMBPO (Thomas et al., 2021), RCPO (Tessler et al., 2018), and LR (Ray et al., 2019).

- **Safe Model-Based Policy Optimization (SMBPO)**: This is a model-based method that uses an ensemble of Gaussian dynamics based transition models. Based on the transition models, it penalizes unsafe trajectories and avoid unsafe states under certain assumptions.

- **Reward Constrained Policy Optimization (RCPO)**: This is a policy gradient method based on penalized reward under safety constraints.

- **Lagrangian relaxation (LR)**: It uses Lagrangian relaxation for the safety constrained RL.

In addition, we have also further extended the proposed RPT with simple data augmentation, producing a data augmented method, RPT+DA, for comparison. Specifically, we perform data augmentation only for the data sampled from the set of risky states $S_U$ in line 14 of Algorithm 1. The MuJoCo environments are built upon robotic simulations, and they take parameters of different parts of the robot as state observations, including coordinates, orientations, angles, velocities, and angular velocities. Each element of the state observation falls into the range of $(-\infty, \infty)$. Hence, for each sampled risky state-action pair $(s_t, a_t)$, we propose to produce an augmented state $\hat{s}_t$ by adding a random Gaussian noise sampled from the standard normal distribution $\mathcal{N}(0, 1)$ to each entry of the observed data $s_t$. We can repeat this process to generate multiple (e.g., $n$) augmented states for each $s_t$. In our experiments, we used $n = 3$. Together with $a_t$, each $\hat{s}_t$ can be used to form an additional risky state-action pair $(\hat{s}_t, a_t)$ for training the contrastive classifier. The hypothesis is that without any prior information about the environment, the training of the proposed contrastive classifier highly depends on the data collected during the agent's interactions with the environment, especially the limited number of observed unsafe states. By using data augmentation, we expect to improve the sample efficiency of the approach while reducing the possible safety violations during the exploration process.

**Implementation Details**   Implementations for LR and RCPO algorithms are adapted from the recovery RL paper (Thananjeyan et al., 2021). For fair comparisons, following the original setting of LR and RCPO, we disable the recovery policy of the recovery RL framework, which collects offline data to pretrain the agent. For MBPO, we adopted the original implementation from the MBPO paper (Thomas et al., 2021). Both SMBPO and RCPO are built on top of the Soft-Actor-Critic (SAC) RL method (Haarnoja et al., 2018). Hence in the experiments we also implemented

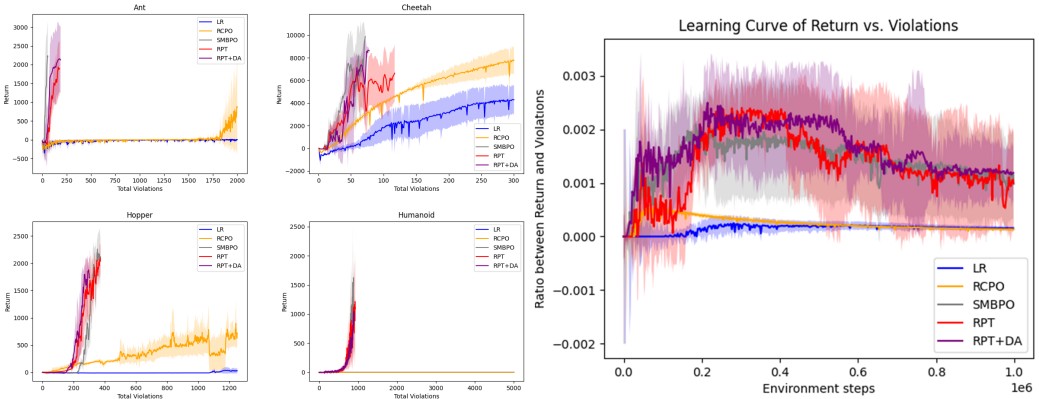

Figure 2: **Left:** For each method, each plot presents the undiscounted return vs. the total number of violations. Results are collected for three runs with random seeds. The curve shows the mean of the return over the three runs, while the shadow shows the standard deviation. **Right:** The plot reports the ratio of the normalized return over the number of violations vs. time-steps. The curve for each method is calculated by averaging across all four environments (Ant, Cheetah, Hopper, Humanoid).

the proposed RPT approach (also RPT+DA) on top of SAC, although RPT is a general safe RL methodology. We used $\eta = 0.9$ for risk preventive trajectory exploration, and collected the mean episode return and violation for $10^6$ time-steps.

## 5.2 EXPERIMENTAL RESULTS

We compared all the five methods by running each method three times in each of the four MuJoCo environments. The performance of each method is evaluated by presenting the corresponding return vs. the total number of violations obtained in the training process. The results for all the methods (LR, RCPO, SMBPO, RPT, and RPT+DA) are presented on the left side of Figure 2, one plot for each robotic simulation environment. The curve for each method shows the learning ability of the RL agent with limited safety violations. From the plots, we can see RPT, RPT+DA and SMBPO achieve large returns with a small number of violations on all the four robotic tasks, and largely outperform the other two methods, RCPO and LR, which have much smaller returns even with large numbers of safety violations. The proposed model-free RPT produces slightly inferior performance than model-based SMBPO on *Ant* and *Cheetah*, where our method requires more examples of unsafe states to yield good performance at the initial training stage. Nevertheless, RPT outperforms SMBPO on both *Hopper* and *Humanoid* with smaller number of safety violations. As a model-free safe RL method, RPT produces an overall comparable performance with the model-based method SMBPO. With data augmentation, RPT+DA further improves the performance of RPT on all the four environments.

To provide a direct and illustrative comparison about the overall performance of the comparison methods across all the four environments, we need to take the average of the results on all four environments using a measure that is independent of any specific environment's return scale. We propose to use the measure that divides the ratio between the return and the number of violations by the maximum trajectory return—i.e., the ratio between the trajectory normalized return and the number of violations, and report the average performance of each method under this measure across all four environments. The results are presented on the right plot of Figure 2. We can see that RPT produces comparable or even better performance on certain region of time-steps than SMBPO, and greatly outperforms RCPO and LR. With data augmentation, RPT+DA produces notable improvements over RPT, especially in the early training phase. The average result curve of RPT+DA is in general above the result curve of the model-based method RCPO across the considered environment time-steps. All these results demonstrate that the proposed RPT approach and its augmented version produce the state-of-the-art performance for safe RL on robotic environments.

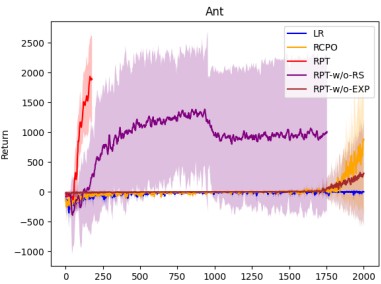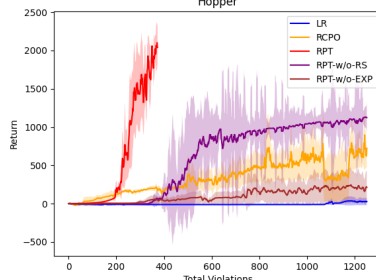

Figure 3: The plot shows the undiscounted return vs. total number of violations, where the purple curve represents RPT without risk preventive reward shaping (RPT-w/o-RS) and the brown curve represents RPT without risk preventive trajectory exploration (RPT-w/o-EXP). Results are collected by the average of three runs. The curve shows mean of the return for the 3 runs, while shadow shows the standard deviation.

### 5.3 Ablation Study

We conducted an ablation study to examine the effectiveness of two main components of the proposed RPT approach, the exploration modification component—Risk Preventive Trajectory Exploration and the reward modification component—Risk Preventive Reward Shaping, on two environments, Ant and Hopper. Specifically, we additionally evaluate the performance of two RPT variants: (1) RPT-w/o-RS, which drops the reward shaping from the full approach RPT by setting $\lambda = 0$; and (2) RPT-w/o-EXP, which drops the risk preventive trajectory exploration component from RPT by removing the risk-prediction based trajectory stopping step on line 12-13 of the training Algorithm 1. Hence, the risk prevention function of RPT-w/o-RS will totally depend on the risk preventive trajectory exploration component, and the risk prevention function of RPT-w/o-EXP will depend on the risk preventive reward shaping component.

We compared these two variants with the full approach RPT and two other model-free safe RL methods, LR and RCPO. The comparison results are reported in Figure 3. We can see that by dropping one component, both RPT-w/o-RS (purple curve) and the RPT-w/o-EXP (brown curve) have dramatic performance degradation comparing with the full approach RPT. This verifies the nontrivial contribution of both components (Risk Preventive Trajectory Exploration and Risk Preventive Reward Shaping) to the proposed approach RPT for safe RL. Meanwhile, the variant RPT-w/o-RS largely outperforms RPT-w/o-EXP and the other two base safe RL methods, LR and RCPO, in terms of the average results, but also demonstrates very large variations over the results of multiple runs, which are particularly notable in the *Ant* environment. This suggests though the risk preventive trajectory exploration component is very useful for safe RL, it is better deployed under a controlled exploration policy under the risk penalized reward. The variant RPT-w/o-EXP produces similar performance as RCPO in the *Ant* environment, and only slightly better performance than LR on both *Ant* and *Hopper*. The results suggest that without the guidance of safe exploration, the effectiveness of reward shaping itself is relatively limited for preventing risks. Overall, the results of this ablation study validate the design of the proposed RPT approach in integrating the strengths of both exploration process modification and reward shaping for safe RL.

### 6 Conclusion

Inspired by the increasing demands for safe exploration of Reinforcement Learning, we proposed a novel mode-free risk preventive training method to perform safe RL by learning a statistical contrastive classifier to predict the probability of a state-action pair leading to unsafe states. Based on risk prediction, we can collect risk preventive trajectories that terminate early without triggering safety constraint violations. Moreover, the predicted risk probabilities are also used as penalties to perform reward shaping for learning safe RL policies, with the goal of maximizing the expected return while minimizing the number of safety violations. We compared the proposed approach with a few state-of-the-art safe RL methods using four robotic simulation environments. The proposed approach demonstrates comparable performance with the state-of-the-art model-based method and outperforms the model-free safe RL methods.

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
