# OpenReview forum: "Safe Reinforcement Learning with Contrastive Risk Prediction"
_ICLR.cc/2023/Conference — Submitted to ICLR 2023_

### Official Review · Reviewer_mSBr · 2022-10-13

**Confidence:** 4
**Correctness:** 3
**Technical Novelty And Significance:** 3
**Empirical Novelty And Significance:** 2
**Recommendation:** 6

**Clarity, Quality, Novelty And Reproducibility:**

The paper is pretty clear and easy to read.

The paper's objective and Theorem 1 is original/novel.

The paper is of above average quality, but figure 1 should be improved.

The paper seems reproducible from the paper descriptions alone.

**Strength And Weaknesses:**

For full disclosure, I have already reviewed a previously submitted (and rejected) version of this paper. My review is based on criticisms/strengths from the old review and old meta-review, along with the changes that have been made to the paper in this new submission. Overall, this paper is much improved from the original version I reviewed.

## Strengths

**Clarity:** The paper is generally clear and readable.

**Mathematical Rigorousness:**

- The contrastive classifier $F_\theta$’s intuition isn’t described very well, but other than that Section 4.1 describing the objective is intuitive and grounded
- The proof of Theorem 1 is well-presented and presents a mostly-grounded way of selecting the dual variable $\lambda$ rather than just treating it as a hyperparameter like in other works.

**Experiments/Results:**

- Experiments and ablations are sensible. The results presented hold up the claim that this works as well as SMBPO while being a model-free method, which is valuable due to fewer assumptions made by model-free algorithms about the environment.

## Weaknesses

**Clarity:**

- Figure 1 is not very useful for demonstrating how the method works. There’s no indication of some risk prediction threshold ($\eta$) nor anything to indicate that actions are actually modified based on this threshold ($a_t, a_t’$ aren’t differentiated).
- The contrastive classifier $F_\theta$’s intuition isn’t described very well, and Eq. 2 just appears without explaining the terms. I understand that this derivation of $F_\theta$ makes Eq 3 convenient (and maybe the authors figured out Eq 3 before figuring out Eq 2), but this could be made more clear.

**Experiments:**

- Assumption 1 is presented without empirical evidence it exists. Can the authors experiment and demonstrate how true Assumption 1 is?
- It’s hard to say that RPT+DA is better, as the authors claim, due to the overlapping standard deviations.

**Method and Framing:**

- Intro: “benefitting from the generalizability of risk prediction, the proposed approach can avoid safety violations much earlier in the training process” → It’s not clear intuitively how this risk prediction differs from the aforementioned prior work at this point in the intro, and therefore how it is more generalizable than prior work.
- Method: “With safety constraints and risk predictions, the proposed approach (to be
elaborated below) is expected to incorporate the strengths of both categories of safe RL techniques.” → by this point, it’s still not clear why this is better. **The authors should make more clear, in the intro and method introduction, that their contrastive method is able to reason about *future risk*, while many current methods either only implicitly do (through reward modifications and TD-learning) or do not at all (binary classification with model-based RL).** This would make the paper's contributions clearer, and make it better in my opinion.

**Summary Of The Paper:**

The authors propose a contrastive-learning inspired objective for shaping the reward function for RL agents to prevent encountering predicted, future “unsafe” states while training. They validate their claims on standard Mujoco environments.

**Summary Of The Review:**

I recommend a weak accept. While I still have some problems with this paper, especially regarding how it's framed, I think the paper is much improved over the previous submission that I reviewed and makes valuable contributions.

---

> ### Author Response · Authors · 2022-11-19
> **Response to Reviewer mSBr**
>
> We thank the reviewer for the detailed review and positive feedback on our new submission. We have taken the comments into consideration and uploaded a revised version of the paper as a supplementary file. Below we provide responses to the questions raised by the reviewer.
>
> **Q1.** Question about the clarity
>
> **Answer:** Thanks for your constructive suggestions. We have modified Figure 1 to make it more illustrative. It now visually shows the relation between our defined threshold $\eta$ with the unsafe region in the environment. We have also added more details to the definition of our contrastive classifier $F_\theta$ in the method section of our revised paper. We use different class-specific weights to define the contrastive classifier $F_\theta$, which is different from the standard Bayes’ classifier.
>
> **Q2.** Question about the experiments.
>
> **Answer:** In the experimental results, RPT+DA is not strictly better, but with the help of data augmentation, the performance of RPT+DA is guaranteed no worse than RPT. In some environments, RPT+DA achieves better performance than RPT. We are still exploring the applicability and generalizability of assumption 1 in different environments. On the other hand, Theorem 1 derived from Assumption 1 only shows the strict correctness of our approach. Practically, standard environments do not require that Assumption 1 strictly holds.
>
> **Q3.** Question about the method and framing.
>
> **Answer:** Previous works focused more on labeling the visited unsafe states and designing of backup policy to avoid the visited unsafe states. In our method, we propose the risk prediction method to avoid the unsafe region before we visit it, which is different from most previous works. We have shown this in the introduction section of the revised version of our paper.

---

> > ### Comment · Reviewer_mSBr · 2022-11-21
> > **Response to author comments**
> >
> > Thanks for the response. I am now satisfied with the quality of Figure 1, and more satisfied regarding the framing in the intro.
> >
> > I think an experiment demonstrating how true assumption 1 is and if there is a correlation between assumption 1 and safeness or environment return would still be useful.
> >
> >
> > After looking at the other reviews and author responses to those reviews, I will be keeping my score the same as I think there are still some unresolved issues regarding clarity. All 3 of us reviewers took issues with the clarity of the paper when introducing $F_\theta$, and I don't think that's addressed yet. Now that the paper revision period is over, perhaps the authors can highlight a detailed plan for how they would make that part clear, addressing all 3 reviewer complaints. Until then I will keep my score the same (still a weak accept).

---

> > > ### Author Response · Authors · 2022-12-04
> > > **Response to Reviewer mSBr**
> > >
> > > We thank the reviewer for the following up discussion. We provide our further response below.
> > >
> > > **Q1.** About experiments on assumption 1:
> > >
> > > **Answer:** Based on the reviewer’s suggestion, we examined assumption 1 on the Ant environment by collecting the risk prediction probability vs. timesteps. The results are provided in the [figure](https://imgur.com/a/BVp63fx), where the Y-axis shows the risk prediction probability and the X-axis represents the timesteps in the unsafe trajectories which are rescaled to the range [0, 1] to unify the length of different trajectories. The blue curve with shadow shows the risk prediction with the increase of timesteps in the trajectories. We also plotted an orange dashed line to show the linearity of the risk prediction curve, which connects the starting risk prediction probability, the risk prediction in the middle of the trajectory, and the ending risk probability.
> > >
> > > We can see that the risk increases with the increase of the timesteps. In the dashed line, the two parts fall within the variance of the blue curve, which means that the predicted risk increases approximately linearly in each part of the trajectory. Thus, assumption 1 strictly holds when the number of timesteps is limited, and the prediction will be more and more precise when the agent is close to the unsafe region.
> > >
> > > **Q2.** About the clarity of the classifier $F_\theta$:
> > >
> > > **Answer:**
> > >
> > > **(1)** It is impractical to directly learn the risk probability $p(y=1|s_t,a_t )$, which is discussed in the first paragraph of section 4.1. We propose to directly learn a binary contrastive classifier $F_\theta$ via contrastive sampling. We **define this contrastive classifier as** $F_\theta=\frac{p(s_t,a_t |y=1)p(y=1)}{(p(s_t,a_t |y=1)p(y=1)+p(s_t,a_t ) )}$ in Eq.(2). The intuition behind this definition is similar to the Noise Contrastive Estimation (NCE) [1,2]. This definition allows one to learn this binary classifier from data by contrastively sampling unsafe-state action pairs as positive samples and sampling general state-action pairs as negative samples. Following Gutmann and Hyvarinen [1, 2], we use a class-specific weight of $p(y=1)$ for positive samples and a weight of 1 for contrastively negative samples. The reason is that we sample the negative samples from the marginal distribution $p(s_t,a_t )$, and hence the negative samples should be weighted $\frac{1}{p(y=1)}$  times more frequent than the positive samples from $p(s_t,a_t |y=1)$. Thus, we define the contrastive classifier $F_\theta$  as
> > >
> > > $$F_\theta=\frac{p(s_t,a_t |y=1)}{p(s_t,a_t |y=1)+\frac{1}{p(y=1)}  p(s_t,a_t ) }=\frac{p(s_t,a_t |y=1)p(y=1)}{p(s_t,a_t |y=1)p(y=1)+p(s_t,a_t ) }$$
> > >
> > > This contrastive sampling strategy **can also be interpreted from the log-likelihood objective function** $L(\theta)$ in Eq.(4). The objective contains the two terms for the positive samples and negative samples respectively, while the positive samples have weight $p(y=1)$ and the negative samples have weight 1.  It is clearly a contrastive sampling-based maximum likelihood training. Below we can show our definition of  $F_\theta$ will achieve the maximization of this objective by making the derivative of $L(\theta)$ wrt $F_\theta$ yield zero:
> > >
> > > $$
> > > \frac{\partial L(\theta)}{\partial F_\theta}
> > > =p(y=1) E_{p(s_t,a_t |y=1)}\frac{1}{F_\theta (s_t,a_t )} - E_{p(s_t,a_t)}   \frac{1}{1-F_\theta (s_t,a_t )}
> > > = E_{(s_t,a_t )} [\frac{p(s_t,a_t |y=1)p(y=1)}{F_\theta (s_t,a_t )}-\frac{p(s_t,a_t )}{1-F_\theta (s_t,a_t )}]=0
> > > $$
> > >
> > > By setting  $\frac{p(s_t,a_t |y=1)p(y=1)}{F_\theta (s_t,a_t)}=\frac{p(s_t,a_t )}{1-F_\theta (s_t,a_t )}$, which makes the derivative above equal 0,  we can derive:
> > >
> > > $$
> > > p(s_t,a_t |y=1)p(y=1) (1-F_\theta (s_t,a_t ))=p(s_t,a_t ) F_\theta (s_t,a_t )
> > > \Longrightarrow p(s_t,a_t |y=1)p(y=1)=[p(s_t,a_t |y=1)p(y=1)+p(s_t,a_t )  ] F_\theta (s_t,a_t )
> > > \Longrightarrow F_\theta (s_t,a_t )=\frac{p(s_t,a_t |y=1)p(y=1)}{p(s_t,a_t |y=1)p(y=1)+p(s_t,a_t )}   (i.e., Eq.(2))
> > > $$
> > >
> > > **(2)** From Eq.(2), we can also easily derive the probability of interest $p(y=1|s_t,a_t )$ in Eq.(3) using the property of Bayes’ theorem:
> > >
> > > $$
> > > p(y=1|s_t,a_t )
> > > =\frac{p(s_t,a_t |y=1)p(y=1)}{p(s_t,a_t )}
> > > =\frac{\frac{p(s_t,a_t |y=1)p(y=1)}{p(s_t,a_t |y=1)p(y=1)+p(s_t,a_t)}}{\frac{p(s_t,a_t)}{p(s_t,a_t |y=1)p(y=1)+p(s_t,a_t )}}
> > > =\frac{F_\theta (s_t,a_t )}{1-F_\theta (s_t,a_t )}
> > > $$
> > >
> > > We will add these derivations along with the references to the Appendix of the paper.
> > >
> > > [1] Gutmann, Michael, and Aapo Hyvärinen. "Noise-contrastive estimation: A new estimation principle for unnormalized statistical models." Proceedings of the thirteenth international conference on artificial intelligence and statistics. JMLR Workshop and Conference Proceedings, 2010.
> > >
> > > [2] A. Mnih and Y. Teh. A fast and simple algorithm for training neural probabilistic language models. ICML2012

---

> > > > ### Comment · Reviewer_mSBr · 2022-12-07
> > > > **REsponse to author's latest comments**
> > > >
> > > > Thank you for adding this. Provided that the authors follow through on their promises of adding both of these things^ to the main paper, I believe this paper should be accepted. I will not change my score solely because I cannot verify that authors will integrate the above clarifications into the paper, however I will argue for acceptance in the reviewer/AC discussion.

---

> ### Comment · Area_Chair_Ncfp · 2022-11-21
> **Any comments to the responses from authors?**
>
> Dear Reviewer mSBr,
>
> Thank you very much for your detailed review.  The authors have provided some clarifications.  How did they change your evaluation?

---

### Official Review · Reviewer_aGao · 2022-10-27

**Confidence:** 4
**Correctness:** 2
**Technical Novelty And Significance:** 2
**Empirical Novelty And Significance:** 2
**Recommendation:** 3

**Clarity, Quality, Novelty And Reproducibility:**

Clarity/Writing Quality

1. The first point at the end of the introduction, is not a contribution but a description of what is being done.
2. Definition of $\tau$ uses $|\tau|$ which leads to a circular definition. You can define $|\tau|$ as the number of actions in the trajectory in a second line.
3. Definition 1 is also partially an assumption since it is assumed that $c$ is binary-valued.
4. Definition 2 is likewise not a definition. It is an algorithmic choice. The true safety is decided by $c$ and not $\eta$ and $p(y=1 \mid s, a)$.
5. Unsafe state set $S_U$ is actually a state-action set
6. What is $\pi$ on line 8? Does it mean $\pi_\phi$
7. What do lines 14 and line 15 do in the Algorithm? Are they being used in optimizing $F_\theta$.
8. $F_\theta$ is the learned model and not the Bayes' classifier. It could be good to separate these two notions.

Suggestions:
1. Consider using $\pi^\star$ as opposed to $\pi^*$

Questions:
1. Can definition 1 be generalized to handle stochastic transitions?

**Strength And Weaknesses:**

Strength:

1. Paper is on safe RL which is an important topic

Weakness:

This paper suffers from three concerns.

1. *Broad Concerns:* The paper makes assumptions that will not apply to applications that the paper uses for motivation in the beginning. To begin with, the paper assumes a deterministic setting that holds in very simple cases. Secondly, a real robotic experiment can immediately fail if the robot hits a wall and breaks down. Therefore, encountering unsafe states even once may not even be an option in such cases. Lastly, $p(y = 1\mid, s, a)$ is a non-stationary distribution that evolves as the policy evolves. This seems undesirable from a learning point of view, as the model has to constantly adjust to the changes in the policy. As the policy optimizes for state-action pairs that are safe/unsafe based on value of $p(y=1 \mid s, a)$ and not directly the value of $c$, therefore, the policy may start taking certain actions or avoid actions, in a given state, as the training progresses. A more desirable strategy would be that $(s, a)$ pairs that are marked as unsafe are never revisited.

2. *Technical Concerns:* Several statements in this paper seem either wrong or unjustified to me. It is my understanding that the classifier $F_\theta(s, a)$ is trained on random $(s, a)$ pairs that are treated as safe $(y=0)$, and a set of unsafe pairs $(y=1)$ that were selected during the previous exploration. If yes, then the Bayes classifier for this problem will be given by $p(y=1 \mid s, a)$ where

$p(y=1 \mid s, a) = \frac{p(s, a \mid y=1) p(y=1)}{p(s, a)} = \frac{p(s, a \mid y=1) p(y=1)}{ p(s, a \mid y=1) p(y=1) + p(s, a \mid y=0) p(y=0)}$.

Here $p(s, a \mid y=0)$ denotes the probability over random state-action pairs in the dataset, as that is the noise distribution for contrastive learning. Now if $F_\theta$ is an expressive class that contains $p(y=1 \mid s, a)$ then we have asymptotically $F_\theta(s, a) \rightarrow p(y = 1 \mid s, a)$ on $(s, a)$ pairs that lies in support of the distribution on which it was trained.

The paper equates $p(y =1 \mid s, a) = \frac{F_\theta(s, a)}{1 - F_\theta(s, a)}$ which does not make sense to me.  Note that in the above version, no correction is required and all probability values are in [0, 1].

Because of this error, or confusion, I am unable to fully check Lemma 1 and Theorem 1.

In another example, assumption 1 seems unclear to me. Does the increase have to be strict? If not, this already rules out deterministic policies since a deterministic policy in a deterministic setting (which is what this paper assumes) will result in a straight path and if this policy visits an unsafe trajectory, then $p(y=1 \mid s, a) = 1$ for all state-action pairs along the way.  This assumption needs to be made clearer to understand it clearly, and an example can help.

3. *Writing Concerns:* The paper is not well-written. I give numerous examples below in the clarity section.

**Summary Of The Paper:**

This paper studies safety in reinforcement learning. The goal is to maximize return while keeping the cost of visiting unsafe state-action pairs below a threshold. The paper proposes learning a classifier $F_\theta(s, a)$ that predicts whether a given state action $(s, a)$ will lead to eventually visiting an unsafe state-action pair. This is then used to update the policy and the $F_\theta$ is then relearned on the fly.

**Summary Of The Review:**

Because of concerns with methodology, writing, and technical statements, I am recommending rejection. My score can change greatly if the authors clarify any misunderstanding. In particular, I would like to know if technical statements concerning the relation between the Bayes' classifier $p(y=1 \mid s, a)$ and the learned classifier $F_\theta$ is true.

---

> ### Author Response · Authors · 2022-11-19
> **Response to Reviewer aGao**
>
> We thank the reviewer for the detailed review and concrete comments. We have taken the comments into consideration and uploaded a revised version of the paper. Below we provide responses to the questions raised by the reviewer.
>
> **Q1.** Question about the broad concerns.
>
> **Answer:** Yes, we adopt the setting that a limited number of violations can be tolerated. As safety violations can never be entirely avoided in real robotic environments, we intend to maximize the performance of the robot while minimizing the number of violations.
>
> The probability that the current state-action pair $(s_t,a_t )$ leads to an unsafe state, $p(y=1|s_t,a_t )$, is partially related to the policy. That is why we also exploit this probability in our safe exploration process. It is not directly related to the value c, but it is a good prediction of the potential vulnerability of reaching such unsafe states. Many previous works have explored labeling $(s_t,a_t )$ as “unsafe states” without making future predictions. However if the agent stops when encountering the “unsafe states”, it will be already too late for the agent to go back to the safe states as it already reaches the “unsafe region” we defined in our paper.
>
> **Q2.** Question about the technical concerns.
>
> **Answer:**
>
> **First,** Eq.(2) is the definition for our proposed binary contrastive classifier $F_\theta$. Note this contrastive classifier $F_\theta$ is NOT derived but defined. It is not equal to $p(y = 1|s_t,a_t)$. We have discussed the reason for not directly learning $p(y = 1|s_t,a_t)$ from data in the first paragraph of section 4.1. Contrastive learning is a concept with various forms. Our definition allows one to learn a binary classifier from data by sampling unsafe state-action pairs from exploration data with weight $p(y=1)$ and using general state-action pairs as contrastive background with weight 1. Similar definitions have been explored in the noise-contrastive estimation (NCE) method [1], where weights for the two parts are specified.
>
> **Second,**  we **derive** $p(y=1|s_t,a_t )$  from $F_\theta$’s definition because we need to compute $p(y=1|s_t,a_t )$ from the classifier’s prediction output. The derivation process is straightforward from $F_\theta$’s definition.
> It is absolutely true that $p(y=1|s_t,a_t )=\frac{p(s_t,a_t |y=1)p(y=1)}{p(s_t,a_t )} =   \frac{p(s_t,a_t |y=1)p(y=1)}{(p(s_t,a_t |y=1)p(y=1)+ p(s_t,a_t |y=0)p(y=0) )}$. But it has no purpose to serve here. We do not need to use this equation to express $p(y=1|s_t,a_t )$ in terms of $F_\theta$’s.
>
> In Assumption 1, we assume the overall probability should increase approximately linearly.
>
> **Q3.** Question about the Clarities.
>
> **Answer:** Thanks for your concrete suggestions. We have fixed most writing issues in our revised paper.
>
> We have added explanations to the first point at the end of the introduction, the definition of $|\tau|$, the typos of $S_U$ and $\pi_\phi$ in our algorithm, and $\pi^\star$ as the optimal policy.
>
> For issues 3 and 4, we follow the literature of safe RL. There are two broadly used settings in safe Reinforcement Learning. In the first setting, an RL agent maximizes its return while satisfying the constraint limit $\sum_{t=0}^T c_t \leq d$, where $c_t$ is given as a cost (normally binary cost) for each state $s_t$. It is used in OpenAI’s safety gym environments [2]. In the second setting, an RL agent stops the trajectory instantly after it reaches a catastrophic state $s_t$, which is broadly used in MuJoCo environments [3]. We formulate this setting in definition 1 and make it consistent with the previous setting. In this setting, all the costs $c_t$ before the end of the trajectory should be 0. A catastrophic state has a cost of 1. Based on this setting, our definition 1 and extended definition 2 should be definitions instead of assumptions.
>
> For issues 7 and 8, line 14 and line 15 are used to update the contrastive classifier $F_\theta$, while the risk prediction of timestep $t$ is written as $p_t$, which is calculated using the trained classifier $F_\theta$.
>
> For the question, definition 1 can be generalized to stochastic transitions if we model the cost function as a distribution of the cost. On the other hand, our approach also works for stochastic environments if we simply feed the cost function as it is determined in the transition $(s_t, a_t, r_t, c_t, s_{t+1})$.
>
> [1] A. Mnih and Y. Teh. A fast and simple algorithm for training neural probabilistic language models. ICML2012
>
> [2] Ray, Alex, Joshua Achiam, and Dario Amodei. "Benchmarking safe exploration in deep reinforcement learning." arXiv preprint arXiv:1910.01708 7 (2019): 1.
>
> [3] Todorov, Emanuel, Tom Erez, and Yuval Tassa. "Mujoco: A physics engine for model-based control." 2012 IEEE/RSJ international conference on intelligent robots and systems. IEEE, 2012.

---

> > ### Author Response · Authors · 2022-12-07
> > **Response to Reviewer aGao**
> >
> > We hope our response has answered the reviewer’s questions and would be happy to answer any follow-up or new questions before the end of the second stage of the discussion phase.

---

> ### Comment · Area_Chair_Ncfp · 2022-11-21
> **Any comments to the responses from authors?**
>
> Dear Reviewer aGao,
>
> Thank you very much for your detailed review.  The authors have provided responses to your concerns.  How did they change your evaluation, particularly on the technical correctness?

---

### Official Review · Reviewer_jp1h · 2022-10-28

**Confidence:** 4
**Correctness:** 3
**Technical Novelty And Significance:** 3
**Empirical Novelty And Significance:** 1
**Recommendation:** 5

**Clarity, Quality, Novelty And Reproducibility:**

**Clarity:** The authors are clear throughout the paper in expressing their motivation and describing their methodology (except for unjustified claims and math in the paper).

**Quality:** The study is a bit behind the quality standard for a venue like ICLR due to my concerns with the credibility of the experiments. See the _Weaknesses_ section.

**Novelty:** In my opinion, the proposed method can be regarded as novel.

**Reproducibility:** I don't believe the reported results are reproducible as the authors did not provide a source code, although they extensively explain the implementation and experimental setup. In fact, the number of seeds used to evaluate the methods' performance is not sufficient.

**Strength And Weaknesses:**

**Strengths:**
- The motivation is clear and promising.
- The prior art is discussed concisely.

**Weaknesses:**
- I couldn't understand some statements in the paper as they seem unjustified. For instance, the classifier $F_{\theta}$ is trained on randomly selected state-action pairs that are treated as "safe" and some pairs that are "unsafe", which are selected in the previous exploration. According to this, the Bayes classifier for such a problem is given by $p(y= 1|s, a)$, which is defined as:
$p(y = 1|s, a) = \frac{p(s, a|y=1)p(y=1)}{p(s, a)} = \frac{p(s,a|y=1)p(y=1)}{p(s, a|y=1)p(y=1) + p(s,a|y=0)p(y=0)}$,

where $p(s,a|y=0)$ is the probability over random state-action pairs in the dataset, corresponding to the noise distribution for contrastive learning. If we consider that $F_{\theta}$ is an expressive class containing $p(y=1|s,a)$, then we asymptotically have $F_{\theta} \rightarrow p(y=1|s,a)$ on $(s,a)$ pairs that lie in the distribution on which the classifier is trained. The paper, however, equates $p(y=1|s,a)=\frac{F_{\theta}(s,a)}{1 - F_{\theta}(s,a)}$, which is left unjustified and I couldn't understand it. Due to such an unjustified claim, Lemma 1 and Theorem 1 become inconsistent to me.

- Why didn't the authors consider giving a continuous "safeness" value between 0 and 1? In my opinion, using a binary representation for the safety of a state-action pair is taking the easy way out. Considering a continuous value, on the other hand, would increase the novelty.
- The comparative evaluations are insufficient as the authors only consider Ant, HalfCheetah, Humanoid, and  Hopper. While these can be considered challenging tasks, I expect the results for BipedalWalker (from BOX2D), Walker2d, and Swimmer. Although the latter has a low state-action dimensionality, it is a very challenging task in which many SOTA methods exhibit poor performance.
- Although it is indicated for the HalfCheetah environment that the robot violates the safety constraint when it falls over, I don't believe it is a credible judgment. Because the Cheetah environment is "stable", a trajectory doesn't terminate unless a pre-specified number of time steps is reached, i.e., 1000. While it'd good to examine the proposed method's performance in a stable environment, the authors don't provide such a discussion.
- The authors should assess a method's performance on a distinct evaluation environment per a pre-specified number of time steps, as it is a common practice in deep RL literature. For example, every 1000 steps, the learning is stopped, and the agent is evaluated in a distinct environment where no exploration or learning is performed for 5-10 episodes. However, the authors used the training rewards as far as I understood.
- The number of random seeds used to produce the reported results is highly insufficient. At least ten random seeds should be used [per the deep RL benchmarking standards](https://ojs.aaai.org/index.php/AAAI/article/view/11694).

**Detailed Comments:**
- An RL agent should be "an RL.." not "a RL".
- Replace the reward function in the MDP definition with $\mathcal{R}$ instead of $r$ as it can confuse readers.
- Punctuation should be put after each equation as they are used in conjunction with the text.
- It looks ugly when a reference is used, such as: "Similar to (Hans et al., 2008)...". Either refer to the natbib package or use it as: "Similar to the work (Hans et al., 2008)...".

**Summary Of The Paper:**

The authors of this study provide a risk-preventive training strategy for safe RL that trains a statistical contrastive classifier to forecast the likelihood that a state action combination would result in unsafe states. They can gather risk prevention trajectories and remodel the reward function with risk penalties based on the expected risk probability to promote safe RL policies. In robotic simulation environments, the authors run experiments. The outcomes demonstrate that the suggested methodology outperforms traditional model-free safe RL approaches and is on par with cutting-edge model-based methods in terms of performance.

**Summary Of The Review:**

While the authors appear clear in their claims and methodology, I don't believe the paper can be accepted to a venue like ICLR. My main concerns with this study are:
1) The simple binary assumption used to define the cost function $c(s, a)$ and unjustified claims.
2) The credibility of the experiments:
- An insufficient number of seeds.
- Insufficient environments.
- Reported returns are training rewards, as far as I understood.
3) Reproducibility as no code or any anonymous repository, which does not reveal the authors' identity, of course, is provided.

---

> ### Author Response · Authors · 2022-11-19
> **Response to Reviewer jp1h**
>
> We thank the reviewer for the constructive comments. We have taken the comments into consideration and uploaded a revised version of the paper. Below we provide responses to the questions raised by the reviewer.
>
> **Q1.** Question about the contrastive classifier $F_\theta$.
>
> **Answer:**
>
> **First,** Eq.(1) is the definition for our proposed binary contrastive classifier $F_\theta$. Note this contrastive classifier $F_\theta$ is NOT derived but defined. It is not equal to $p(y = 1|s_t,a_t)$. We have discussed the reason for not directly learning $p(y = 1|s_t,a_t)$ from data in the first paragraph of section 4.1. Contrastive learning is a concept with various forms. Our definition allows one to learn a binary classifier from data by sampling unsafe state-action pairs from exploration data with weight $p(y=1)$, and using general state-action pairs as contrastive background with weight 1. Similar definitions have been explored in the noise-contrastive estimation (NCE) method [1], where weights for the two parts are specified.
>
> **Second,**  we **derive** $p(y=1|s_t,a_t )$  from $F_\theta$’s definition because we need to compute  $p(y=1|s_t,a_t )$  from the classifier’s prediction output. The derivation process is straightforward.
> It is absolutely true that $p(y=1|s_t,a_t )=\frac{p(s_t,a_t |y=1)p(y=1)}{p(s_t,a_t )} = \frac{p(s_t,a_t |y=1)p(y=1)}{(p(s_t,a_t |y=1)p(y=1)+ p(s_t,a_t |y=0)p(y=0) )}$. But it has no purpose to serve here. We do not need to use this equation to express $p(y=1|s_t,a_t )$ in term of $F_\theta$’s.
>
> **Q2.** Questions about continuous “safeness”.
>
> **Answer:** In our setting, we follow the safety constraints defined in MuJoCo such that the RL agent is given a discrete cost of 1 when it violates the safety constraint, otherwise it receives a cost of 0. Because the cost for each timestep is discrete, we intend to derive a continuous “safeness” of each state for prediction purposes. In our work, we design the contrastive classifier $F_\theta$ to predict the risk of the current state leading to unsafe states, which is considered as a continuous “safeness” that falls in the range of $[0, 1]$. Based on the predicted risk, we aim to train the agent safely with safe exploration and reward shaping.
>
> **Q3.** Questions about evaluation in the experiments:
>
> **Answer:** We follow the evaluation metrics adopted in the SMBPO paper for a fair comparison. The return is reported on the evaluation environment after every 1000 timesteps of training. We used the four environments (Ant, Cheetah, Humanoid, and Hopper) for two reasons: First, they are challenging tasks used in safe RL. Second, they have been used as the evaluation environments in previous works (LR, RCPO, and SMBPO). We adopted them to achieve fair comparisons. We did not take other tasks as they are not presented in the related works we compare with. In our current experiments, we used the average of 3 runs the results already achieve the comparison purpose with acceptable variance. But sure we will add more runs into the results.
>
> **Q4.** Question about the Cheetah environment:
>
> **Answer:** Thanks for pointing out the confusion. Instead of using the HalfCheetah environment, we adopted the modified Cheetah environment defined in the work of SMBPO [2] with extra safety constraints. In the standard HalfCheetah environment, the robot stops with a violation when it reaches a fixed limit of maximum trajectory length of 1000. In the modified Cheetah environment, though the RL agent normally reaches the maximum trajectory length, it also stops and receives a safety violation when the robot’s head flips on the ground. We have clarified this in the revised paper. We have also revised the paper to fix the other minor issues.
>
> [1] A. Mnih and Y. Teh. A fast and simple algorithm for training neural probabilistic language models. ICML2012
>
> [2] Thomas, Garrett, Yuping Luo, and Tengyu Ma. "Safe reinforcement learning by imagining the near future." Advances in Neural Information Processing Systems 34 (2021): 13859-13869.

---

> > ### Comment · Reviewer_jp1h · 2022-11-21
> > **I'd like to thank the authors for their response**
> >
> > I'd like to thank the authors for addressing my concerns. My responses to the authors' rebuttal can be summarized as:
> >
> > **Derivation of the contrastive classifier $F_{\theta}$:** OK, as far as I'm concerned. But please insert a clarification regarding this issue.
> >
> > **"Safeness":** It's fine.
> >
> > **"Stableness" of the HalfCheetah environment:** If the authors used the modified version, it's OK.
> >
> > **Evaluation in the experiments:** The authors stated that "_the average of 3 runs the results already achieve the comparison purpose with acceptable variance_", how do know the value for an **acceptable variance**? More seeds should've been used. If there is a reference, please provide it. I can change my score.

---

> > > ### Author Response · Authors · 2022-12-04
> > > **Response to Reviewer jp1h**
> > >
> > > We thank the reviewer for the following up discussion.
> > >
> > > **About the evaluation in the experiments:**  Reporting results on the average of 3 runs (seeds) has been adopted in many existing works. For example, the introductive paper to OpenAI gym [1] reports their metrics over the average of 3 runs; the recovery RL [2] evaluates their method with 3 random seeds in the physical environments. RCPO [3] does not report the number of seeds they used in their paper. With 3 runs, the variance is acceptable because more runs does not change the comparison relationships of the curves for different approaches. Nevertheless, we agree that experiments with more runs (seeds) will definitely reduce the variance and stabilize the results. We will add more runs in the final version of the paper.
> > >
> > > [1] Ray, Alex, Joshua Achiam, and Dario Amodei. "Benchmarking safe exploration in deep reinforcement learning." arXiv preprint arXiv:1910.01708 7 (2019): 1.
> > >
> > > [2] Thananjeyan, Brijen, et al. "Recovery rl: Safe reinforcement learning with learned recovery zones." IEEE Robotics and Automation Letters 6.3 (2021): 4915-4922.
> > >
> > > [3] Tessler, Chen, Daniel J. Mankowitz, and Shie Mannor. "Reward constrained policy optimization." arXiv preprint arXiv:1805.11074 (2018).

---

> ### Comment · Area_Chair_Ncfp · 2022-11-21
> **Any comments to the responses from authors?**
>
> Dear Reviewer jp1h,
>
> Thank you very much for your detailed review.  The authors have provided responses to your concerns.  How did they change your evaluation, particularly on the insufficiency of experiments.

---

### Decision · Program_Chairs · 2023-01-20

**Decision:**

Reject

**Justification For Why Not Higher Score:**

There are inconsistencies in the derivation or theoretical support of the proposed method, and it is unclear why and when the proposed approach works.

**Justification For Why Not Lower Score:**

N/A

**Metareview: Summary, Strengths And Weaknesses:**

This paper proposes a method of safe reinforcement learning (RL) that learns a classifier for distinguishing between safe state-action pairs from unsafe state-action pairs.  A novelty is in contrastive learning of the classifier to enable stable learning.  The effectiveness of the proposed approach is well supported by the experiments with the safety gym environments, which constitute the major strength of the paper.

The major weakness is in the derivation or the theoretical support of the proposed approach.  While some of the the initial confusions among the reviewers about the definitions and the use of F_\theta are resolved through the discussion with authors and among reviewers, there remains questions and potential improves that should be made around the expositions or definitions of F_\theta or p(y=1|s,a).


*The following is a personal opinion of the area chair:

Consider the behavior policy that eventually reaches an unsafe state almost surely.  The behavior policy considered in Section 5.1 is such a policy ("RL agent reaches the end of the trajectory once it encounters the safety violation").  Then p( y=1 | s, a ) = 1 for any (s, a). Then, F_theta(s,a)=1, Assumption 1 cannot hold, etc.  Despite this, in the feedback from the authors, Assumption 1 is shown to be empirically satisfied to some extent in one environment, and overall the proposed approach is working fine (surprisingly given the inconsistency in the theory). Perhaps p( y=1 | s, a ) is representing something different, but it is unclear what it is.  So, the approach appears to be working, but its derivation or its theoretical support are confusing.  Due to these inconsistencies in theory, it is unclear why and when the proposed approach works.